# Toto-Cell: A new software to analyze cellular events during video-microscopy

**Léa-Isabelle Renaud**[1], **Christophe Renaud**[2], **Samuel Delepoulle**[2], **Eric Asselin**[1]*

**1** Laboratoire de Gynéco-Oncologie Moléculaire, Université du Québec Trois-Rivières, Trois Rivières, Québec, Canada, **2** Laboratoire d'Informatique, Signal et Image (LISIC), Unfigiversité du Littoral Côte d'Opale, Calais, France

☯ All these authors are contributed equally to this work.
* eric.asselin@uqtr.ca

**Data Availability Statement:** All relevant data are available at https://gitlab.com/crrenaud/cellv1.5.

**Funding:** This work was supported by the Natural Sciences and Engineering Research Council of Canada (NSERC) (2019-06151 to EA). The funders

## Abstract

Video-microscopy is a technology widely used to follow, in a single cell manner, cell behavior. A number of new studies are searching a way to track these behaviors by artificial intelligence; unfortunately some real-time events still have to be track manually. For that reason, we developed a software that helps the experimenter to analyze collected data. Toto-cell is very simple to use and it can be adapted at different type of analyses or treatments. It allows a wide new range of parameters that were nearly impossible to calculate only by hand. We thus developed this new software using HEC-1-A endometrial cell line to track different cellular parameters such as: the number of normal/abnormal mitosis, the ratio per day of death, mitosis, cell fusions or finally the length between two mitosis cycles. We treated our cells with cisplatin, doxorubicin or AZD5363 (an Akt inhibitor) to obtain different cellular events. What emerged is a huge heterogeneity for these analyzed parameters between the cells in a single treatment which is clearly demonstrated by the results provided by Toto-Cell. In conclusion, our software is an important tool to facilitate the analysis of video-microscopy, in a quantifying and qualifying manner. It enables a higher accuracy when compared to manual calculations.

## Introduction

Video-microscopy may appear to be a new technology but in fact it has been used since 1976 with the C.Elegans model [1]. The principle of this imaging technique is to follow the different behaviors of animated living things by taking pictures at a given latency and by making videos with these pictures. In 1976, scientists took their own images manually and had to stayed close to the microscope during the entire experiment. In fact, since then, improvements in terms of automation have been developed: our microscope's cameras are now able to take pictures in real-time. Nowadays the next more important problem is always the analysis of the videos and pictures. A lot of new projects are meant to improve the analysis of these videos [2], but due to the diversity in cell morphology and types, differences between imaging protocols (use of a confocal, an Incucyte, or a home-made installation for examples), it is really difficult to obtain

had no role in study design, data collection and anal-ysis, decision to publish, or preparation of the manuscript.

**Competing interests:** The authors have declared that no competing interests exist.

a reliable automation of image analysis. Creation of a way to detect different events in time-lapse videos is currently in progress, but most of these projects are for normal mitosis model [3], which means a fully symmetric division into two cells. Somehow the other events such as mitosis in more than two or fusion appear difficult to be taken into account for the moment. This means that further manual analysis is required in videos containing this type of complex behavior, which frequently occurs in cancer cells.

In biology, a new area of single cell analysis has changed our knowledge of cancer therapy. Single cell sequencing in genomics or proteomics leads to a precise way of analyzing the molecular pathways inside tumor cells [4, 5]. Thus, video-microscopy can be used as a single cell technique in cancer biology; conversely to the current methods that are only taking a snap-shot only at the end of the experiment [6], video-microscopy can show the different cells behaviors consistently during the entire experiment. In testing new cell drugs, it is useful to know if the treatment works or not and moreover to elucidate what kind of events are arising. In fact, previous methods such as clonogenic assay and scratch assay can be linked to video-microscopy to clarify if a treatment is working for example by killing the cells, or by stopping mitosis for an amount of time. According to this, heterogeneity in cancer is the most common point of therapy failure [7], each individual cell being able to react in its own way to a treat-ment. Without single cell analysis, it is impossible to link the individual behavior of one cell to its fate at the end of the experiment. With this same idea in mind, we observed in our videos some events called fusions. This event is not a well-known and described phenomena, and is characterized by the fusion of two or more cells into one. It is unclear for now, how fusions on cancer initiation and progression are implicated but what we know is that the cell fusion fre-quency is increasing with the tumor size [8]. As expected the fusion of cells increases the tumor heterogeneity where genomic instability and chromosomal variations play a critical role in driving the evolution of tumors [9]. For all these reasons, video-microscopy are required to be annotated manually to better understand how a treatment works at the single cell level.

The main aim of this article is to present the Toto-Cell software developed by our team. This tool was created to facilitate the graphical presentation of the manual annotation of a video-microscopy and to optimize its analysis phase. We performed video-microscopy experi-ments with different types of drugs and showed parameters that can be useful to analyze the cell's behavior after a treatment. This article shows the different parameters that Toto-Cell can extract from the data, how it calculates them and its application on endometrial cancer cells.

## Materials and methods

### Cell culture and treatments

Human endometrial adenocarcinoma cell line HEC-1-A (from ATCC) were cultured respec-tively in Mc Coy medium supplemented with 5% of BGS, with 0.1% gentamycin. Cells were grown in an incubator at 37°C and 5% $CO_2$. Cells were cultured for no longer than 10 weeks.

Cells were treated with anti-cancerous treatments at different concentrations: Cisplatin (Selleckchem, NSC 119875) 10$\mu M$, Doxorubicin (Selleckchem, S1208) 3$\mu M$, and AZD5363 (Adooq, A11759) 20$\mu M$. Control was defined by adding the same volumes of diluant (respec-tively saline for Cisplatin, DMSO for Doxorubicin and AZD5363).

### Video-microscopy

The experimental scheme is described in Fig 1. Namely, 24 h before treatment, the cells were plated in a 8-wells rack (Thermofisher—12-565-338) 15 000/$cm^2$. 24 h after treatment, media was removed, cells were washed carefully with PBS, and 500$\mu L$ of media was added with a dilu-tion of 0.5$\mu M$ of SiR- DNA (SPY650-Spirochrome) to counterstain the nucleus of alive cells.

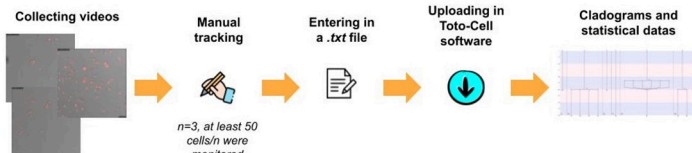

**Fig 1. Temporality of video-microscopy.** Cells were plated 48,5 hours before the start of the video-microscopy; 24 hours after the plating, cells were treated with controls, chemotherapy or targeted treatment, the Sir-DNA was added 30min before the start of the experiment. After that, one picture in two channels (brightfield and far red) were taken each 10min during 72 hours. At the end of the 3rd day, the pictures were compiled into a movie.

Then, the rack was inserted into an incubation chamber (INUB-GSI2F1, Tokai Hit) and then imaging was performed with a Leica TCS SP8 confocal microscope. After setting up standard cell culture conditions (i.e., 37°C/5% $CO_2$ and water), pictures of 10 to 20 cells were taken at 20X (Leica microsystems) magnification at 3 separate locations in each well every 10 min over 3 days on 2 different channels: a phase contrast brightfield channel to obtain the whole cell morphology and a far-red fluorescence channel (λabs/Em 664/684nm) to detect SirDNA-labelled nuclei. The 2 channels allowed the suitable tracking of cellular observables. Pictures were taken at 1024x1024 format, speed 200, zoom factor 1 and at magnification 20x/0.75 DRY (HC PL APO CS2). Complete movies were obtained from the compilation of the 432 images from both channels. Movies were transformed into .tif files using the Leica Application Suite (Leica, Advanced Fluorescence, version 4.0.0.11706). Each cell was then manually tracked frame by frame over the 432 frames using Fiji visualization tool [10]. The type and timing of appearance of phenotypic observable (cell count, normal mitosis, non-dichotomic mitosis, cell fusion, mitotic death, non-mitotic death) were manually monitored (Fig 2). Abnormal mitosis was characterized by either a division in more than 2 cells (non-dichotomic mitosis) or normal mitosis followed by the coalescence of the daughter cells (cell fusion). Cell death was characterized as a function of the presence or not of a previous cell division (mitotic death and non-mitotic death were identified in a similar manner).

## Analysis of cell monitoring data

The results of the manual tracking are then provided to our Toto-cell software. This application has been developed to graphically represent the evolution of the cells studied in a video-microscopy experiment, as well as providing a range of statistics concerning this evolution. It was developed in the C++ language and can be used via a Web interface for ease of use (see Fig 3).

The application takes as input a file in which the person who manually analyzed the video has noted the evolution of each of the cells studied in the experiment. After analyzing the input file, it produces a graphical representation, in the form of cladograms of the evolution of all the

**Fig 2. Toto-Cell bio-statistical processing of video-microscopy.** Each video was collected and tracked manually. At least 50 cells were followed for one repetition, for a mean of 150 cells per condition. All the data were entered into a .txt file thas was then uploaded into Toto-Cell. It gave us cladograms and a .csv file with our statistical datas.

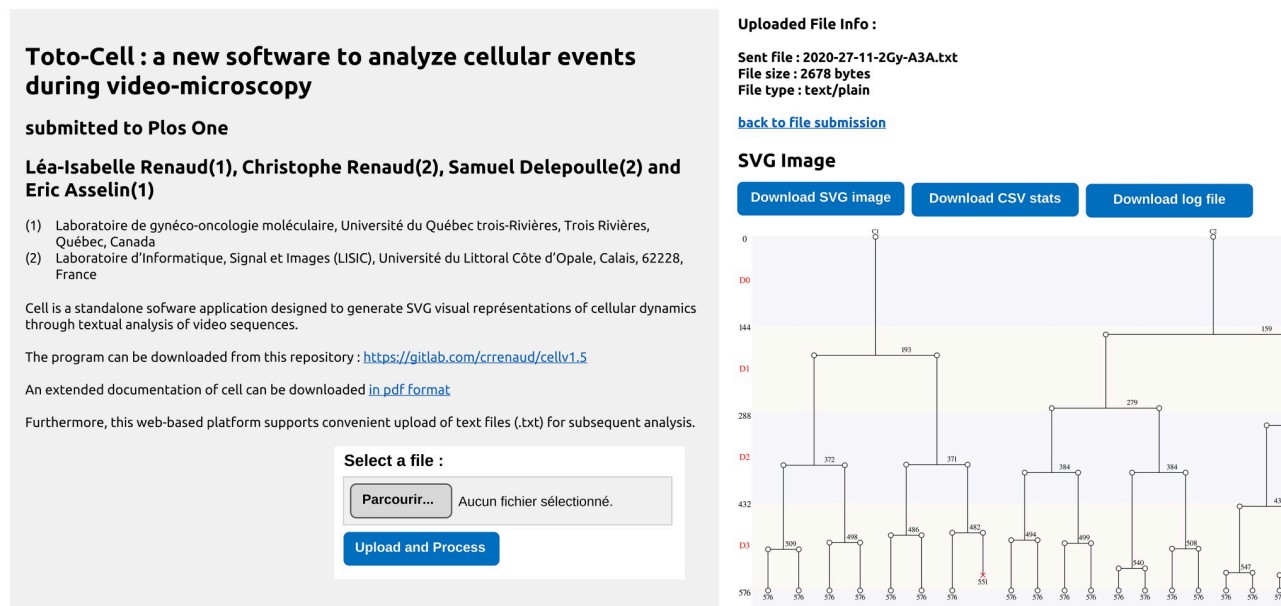

**Fig 3. The web interface of Toto-Cell.** The web interface offers to load an event file available on the experimenter's computer (a). Once the file has been selected and uploaded, Toto-Cell performs the analysis remotely, displaying the results on a second page (b). The user can then retrieve the image representing the cladogram and the statistics file, and return to the submission page if necessary. The submission page can be accessed at https://diran. univ-littoral.fr/web-cell/cell.html.

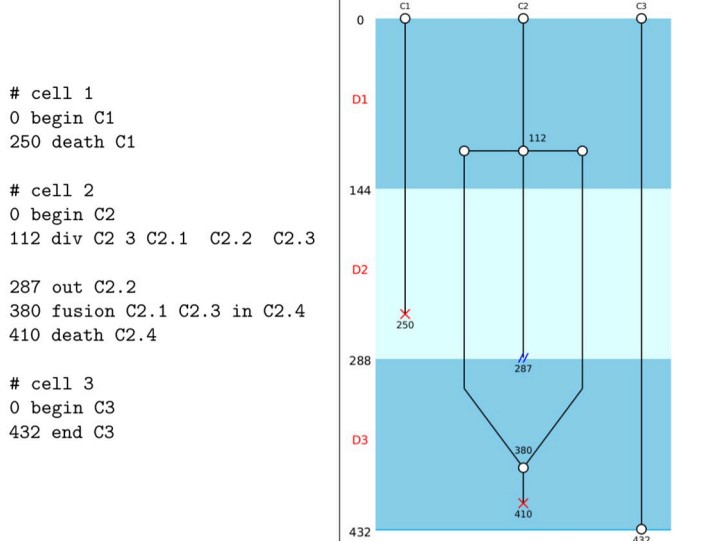

**Fig 4. Example of an event file.** 3 cells were analyzed and the events relating to them were reported in the text file whose content is shown on the left: the cells are all present at the start of the video (time 0) and are named C1, C2 and C3 at the time they were created (`0 begin C1` for example). Cell C1 dies at time 250, while cell C2 undergoes abnormal mitosis into 3 cells at time 112. These 3 cells are named (C2.1, C2.2, C2.3), with the second cell leaving the camera field at time 287. The daughter cells C2.1 and C2.3 then fuse at time 380 before the new cell obtained (named C2.4) dies at time 410. Finally, cell C3 is not affected by any event and we mark the end of the analysis at time 432 (end of the video). The right part of the figure present the corresponding cladogram.

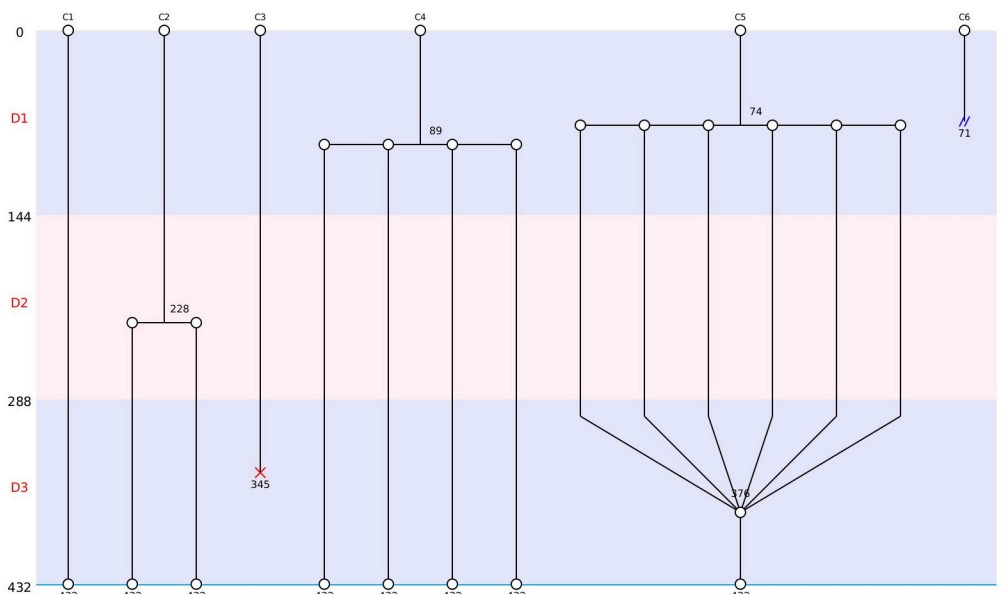

**Fig 5. Representation of the different events by Toto-Cell in cladrogramm.** This cladrogramm shows the different events that Toto-Cell can produce. At the left side, the days and their definition by frames, in the top the name of the cells. The C1's cell has no event during the 3 days; the C2's cell has a division in the frame 228; the C3's cell dies in the 345th frame; the C4's cell is dividing in 4 in the frame 89; the C5's cell is dividing in 6 daughter cells in frame 74 and fusion all together in 376; finally the C6's cell is getting out the field at frame 71.

cells studied, as well as a file of statistics on this evolution. The graphical representation is generated in the form of an image in SVG graphics format. This format is recognized by most image-editing software and offers the possibility to zoom in on different parts of the image without any loss of quality. This is a real advantage when the number of cells in the video sequence grows. The statistics file is generated in CSV format, which is recognized by most spreadsheet programs, which makes them easier to use.

The video sequence analysis file must be created by the person analyzing the video in a text file, following very simple and intuitive instructions relating to each of the events that can be encountered in a video: presence of a cell, death or exit of a cell, division into several cells or fusion from several cells present. In order to identify each cell processed, the experimenter must assign them a name, which must be unique for each video. Each instruction should be indicated in the same way, by first indicating the number of the image in which it occurs, followed by a keyword corresponding to the event identified and followed by the name of one or more cells affected by this event. An example of such a file, associated with the image obtained, is provided in Fig 4 and the documentation available on the Toto-Cell web page details all the functions available in the software.

Note that a version of Toto-Cell is available for installation directly on the user's machine, which can be useful if there is no network connection, for example. A version of the software exists for Windows, MacOs and Linux on the gitlab software repository (https://gitlab.com/crrenaud/cellv1.5), where the application sources and documentation can also be found.

## Figures and charts

Figures were made with GraphPad Prism v9.5 and charts with Xmind v23.11. All the cladograms were generated by Toto-Cell.

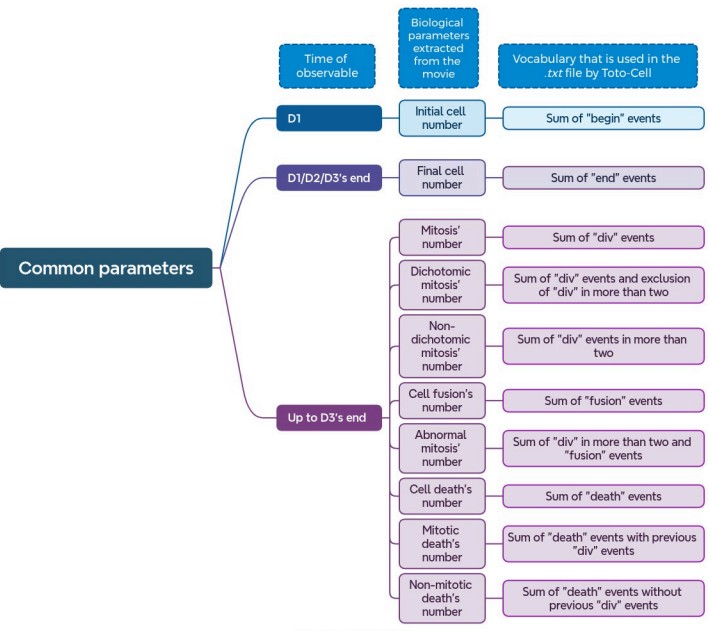

**Fig 6. Common parameters extracted from video-microscopy data by Toto-Cell.** This chart is showing in three different columns the time of observable, the biological parameters that are extracted from the videos and the vocabulary used in the .txt file that Toto-Cell translates into statistics. Briefly, initial cell number is an indicator of how many cells have the event "begin", it permits to show how many cell the experimenter has followed. The final cell number is the accounting of all the cells that finished at the end of the video, it is translated by an "end" event in the .txt file. Mitosis number is the sum of all the "div" event either abnormal or normal divisions. Dichotomic and non-dichotomic mitosis' number are respectively the exclusion of one and the other. That is to say, dichotomic mitosis count do not take into account divisions in more than two, and non-dichotomic mitosis count do not take into account normal mitosis. Next, the fusion number is a number calculation of fusions in the video. For abnormal mitosis number it is the account of each mitosis that was abnormal or followed by a fusion. And finally, a characterization of the death, with the total number of cell death (sum of "death" events) and if it was preceded of a mitosis.

## Results

Different events can occur in video-microscopy and are treated by Toto-Cell: division (normal and abnormal), fusion, death and an exit of the field. We used here Hec-1-a endometrial cancer cells treated in monotherapy, filmed with confocal imaging technology, to test our software. For our experiment, 3 days are 432 frames. To help the experimenter with the analyses of these events we defined parameters on Toto-Cell.

### What produces Toto-Cell?

**Phylogenic trees or cladogramms.** As explained before, the first cladograms were designed in the 70's with C.Elegans. The nomenclature is still the same since this period. As you can see in Fig 5, according to the methodology a separation and new branches correspond to a division and a red cross means death. In our representation, a double parallel blue bar stands for an exit out the field. A fusion is characterized by the union of a number of cells. If a cell has spent all the time of the video-microscopy without any event, it would be represented by a straight line from the top to the bottom of the cladogram. At the top of the cladogram, the user can find the cells' names. Here we used a number for defining the cells. At the left of the cladogram you can find the days and the number of frames that composed a day. For each

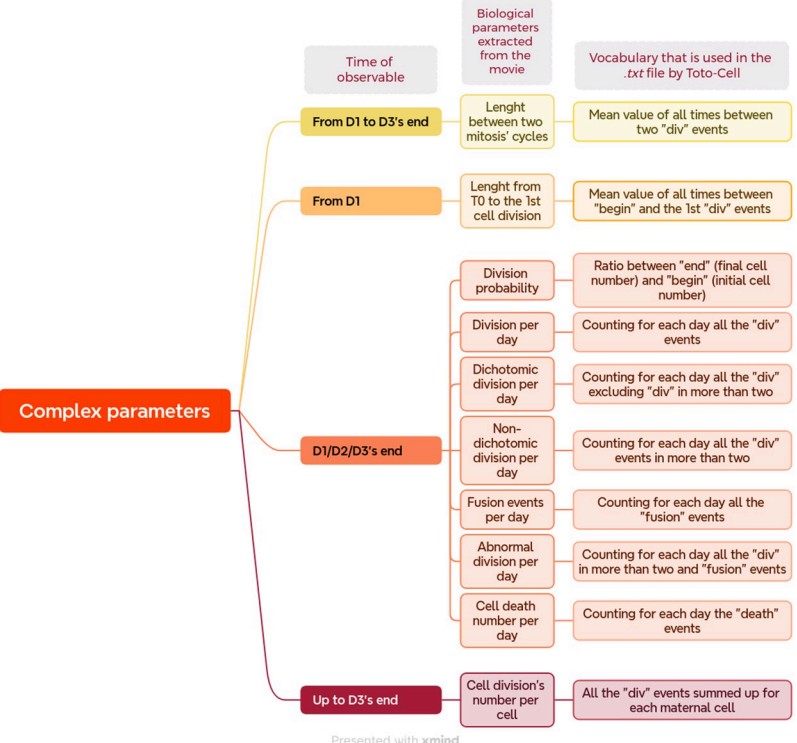

**Fig 7. Complex parameters extracted from video-microscopy data by Toto-Cell.** This chart is showing in three different columns the time of observable, the biological parameters that are extracted from the videos and the vocabulary used in the .txt file that Toto-Cell translates into statistics. These parameters are using mostly the previous one in Fig 6 to do calculations and have statistics. Firstly, the length between two mitosis' cycle is calculated by the mean of the time between two "div" events for all the cells that were followed. The length between T0 (start of the video microscopy, "begin" event in the .txt file) and the first event "div", mean value of all the followed cells that were followed. The division probability is a ratio between the number of cells that have an "end" and a "begin" events, it excludes all the cells that died, or went out of the field. To pursue, the others parameters are the different events but in ratio per day. And lastly, we have the cell's division number per cell, which summarizes all the "div" events for one maternal cell.

event, the experimenter can see the event and the event frame number, which is always below the event.

**Quantifying and qualifying data.** Toto-Cell parameters are divided into two classes: the first ones are common parameters shown in Fig 6 and the others are the complex ones in Fig 7. These parameters are directly calculated from the data that are on the .txt file. We have quantifying parameters to basically help to understand in a numeric manner the treatment's effect. In Fig 6, initial cell number is a parameter that allows the experimenter to check the right amount of cells wanted. It calculates only the number of cells that have the event "begin" in the .txt file. The final cell number considers all the cells that finish at the end of the experimentation, not taking into account if it is a mother cell or a daughter cell, or the other events that can occur during the time of the video-microscopy.

Qualifying parameters are helping the characterization of different events, such as mitosis or death. The mitosis number summarizes all the divisions no matter the quality of the mitosis. More precisely we develop 3 tools to help qualify mitosis. First, we have the cell fusion's number that is the sum of the "fusion" events; these fusions can be between daughter cells after a division or unexpectedly with another cell with no previous lineage with the first one.

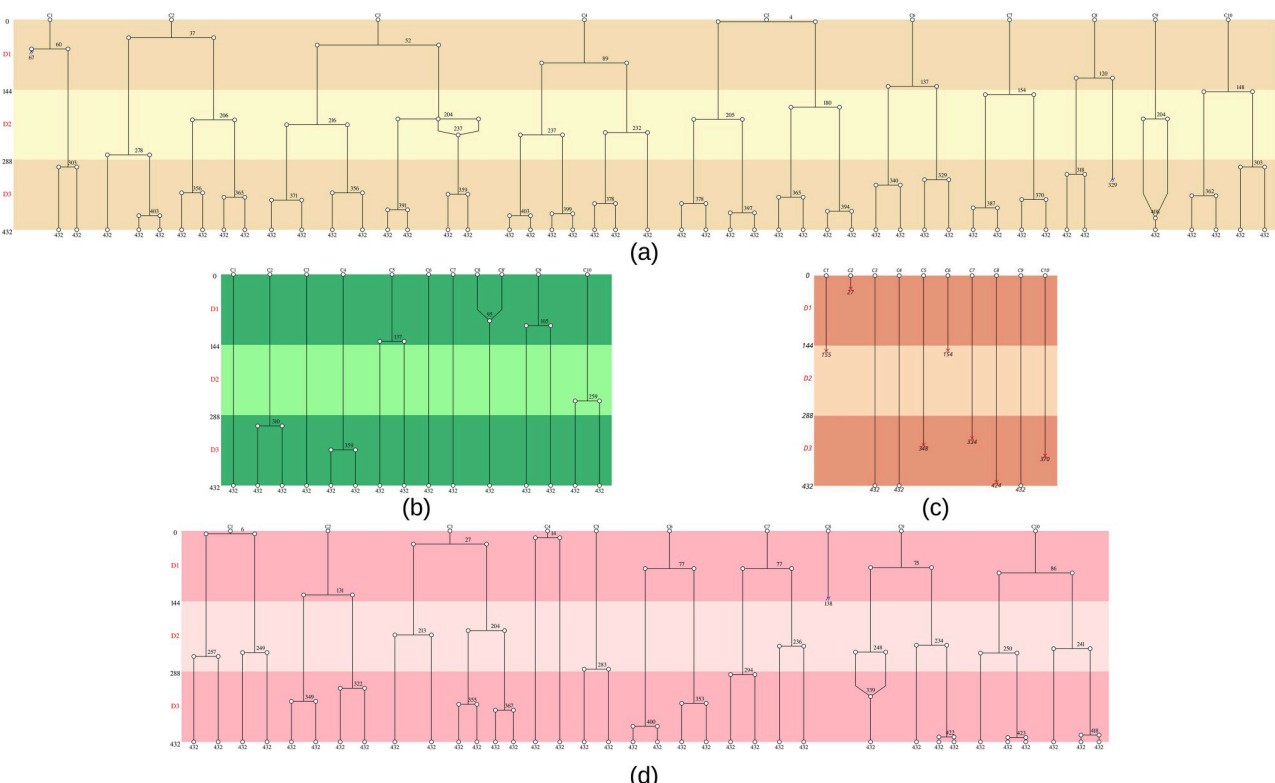

**Fig 8. Cladograms representing treated (24h) HEC-1-A cells' behaviors. (a)** Control cells, **(b)** Cisplatin treated cells ($10\mu M$), **(c)** Doxorubicin ($3\mu M$) treated cells, **(d)** AZD5363 treated cells ($20\mu M$)—three independent experiments.

Secondly, we develop a counter of non-dichotomic mitosis' number, which means that Toto-Cell is counting all the "div" events that are in more than two as non-dichotomic. Finally, we have an abnormal mitosis counter, this one being defined as the mitosis in more than two and the fusion that can occur after a mitosis. Deeply, Toto-Cell is taking into account as "abnormal mitosis" only a mitosis followed by one of these two events; for example if a cell is doing a division in 4 and after that a fusion, Toto-Cell counts only one event. For death, we develop a tally, and precise after that the type between non-mitotic and mitotic death also known as a death following mitosis. We decided to study the length between two mitosis cycles: how long does it take to a cell to re-prepare itself to a new mitosis. It is a good indicator because a longer/shorter time for preparing cell's division can be explained by damages into the nucleus and chromosomes. This parameter is taking into account all the cells from the start to the end of the video. We also measure the time between the start of the video and the first division; this allows us to see if the treatment has delayed time of the first division.

To go further, we have defined "complex parameters" (see Fig 7) which require Toto-Cell to perform a number of calculations.

Conversely to the previous one that can potentially be calculated by hand without wasting too much time, these one would be too tedious and error-prone on such a large sample. Among them, we decided to study the length between two mitosis cycles: how long does it take to a cell to re-prepare itself to a new mitosis. This parameter is taking into account all the cells from the start to the end of the video. In the same line of quantification, we decided to take all the previous common parameters and to do a ratio per day, it allows us to see how and if the treatment has an effect, either delayed, over the long term, or, for example, directly but over

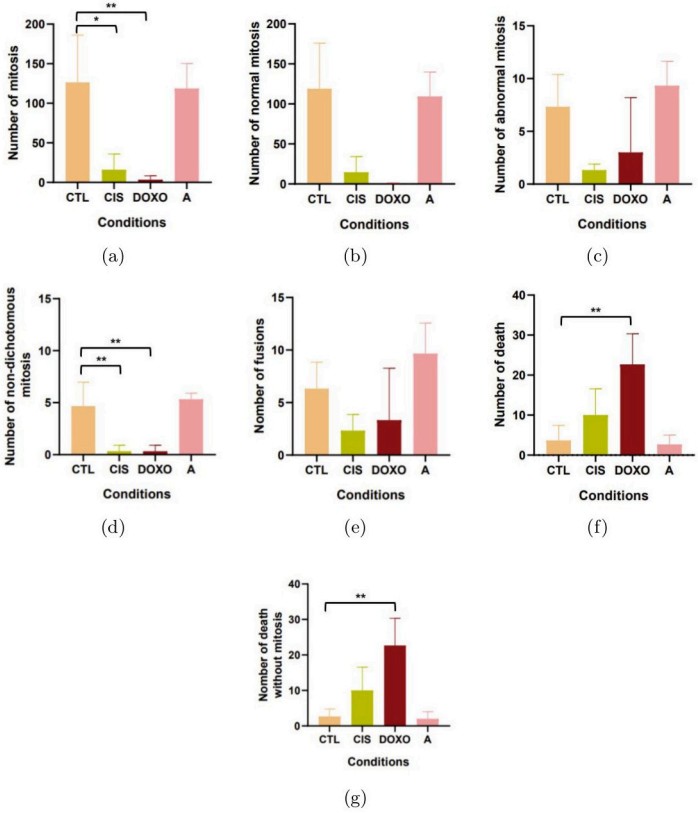

**Fig 9. Application of simple parameters on control and treated (24h) HEC-1-A cells.** CTL: control condition, CIS: cisplatin condition ($10\mu M$), DOXO: Doxorubicin condition ($3\mu M$), A: AZD5363 condition ($20\mu M$). **(a)** Number of mitosis on 3 days, **(b)** number of abnormal mitosis on 3 days, **(c)** number of abnormal mitosis on 3 days, **(d)** number of non-dichotomous mitosis, **(e)** number of fusions on 3 days, **(f)** number of death on 3 days, **(g)** number of death without mitosis. Anova one way followed by Dunett's test, results are mean ± S.D of three independent experiments. *=$P < 0.05$; and **=$P < 0.01$.

the short term. We also have a ratio between final cell number and initial cell number that enables to see the division probability on the total days of the experiments. Finally, depending on the heterogeneity of the cells responding to a treatment, we are developing a counter of events per cell, with which the experimenter can see whether a cell stands out from the others and study it in greater depth.

## An example of how Toto-Cell works

We decided to test our software on data from video-microscopy. We used our Hec-1-a cells treated with cisplatin, doxorubicin or AZD5363 (an Akt inhibitor). The first two are chemotherapies that are used commonly in clinics for endometrial cancer, and the last one is a targeted therapy used in clinical trials [11]. We followed 150 cells for each condition.

**Cladograms.** First we looked at the cladograms produced by Toto-Cell (Fig 8).

For space reasons, here we put an extracted representative cladogram of 10 cells but the entire cladograms (for one repetition, 50 cells) can be found on the following additional webpage, with associated data (cell evolution and statistics): https://www.lisic.univ-littoral.fr/~renaud/VideoMicroscopy/PlosOne/.

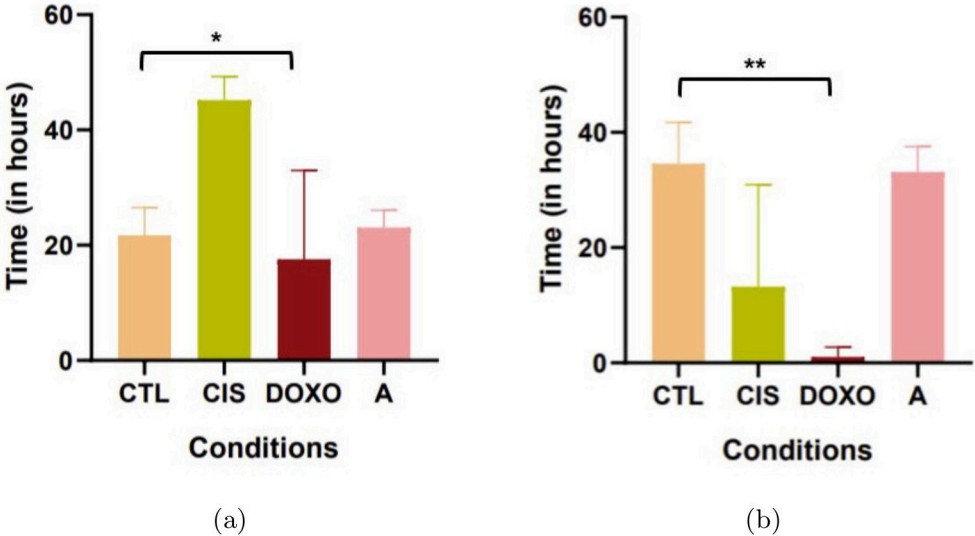

**Fig 10. Representation of length of mitosis on control and treated (24h) HEC-1-A cells.** CTL: control condition, CIS: cisplatin condition ($10\mu M$), DOXO: Doxorubicin condition ($3\mu M$), A: AZD5363 condition ($20\mu M$). **(a)** Mean of time between two mitosis (on 150 cells by condition) and **(b)** between D0 (start of the video-microscopy) and the first division (on 150 cells by condition). Anova one way and Dunett's test, results are mean ± S.D of three independent experiments. *=$P < 0.05$; and **=$P < 0.01$.

In control cells (Fig 8a), the behavior's pattern was really repetitive, one division occured each day. It was totally different for the cells treated with chemotherapies. With cisplatin treatment (Fig 8b) the cell divisions appeared to be delayed or have disappeared. For doxorubicin treatment (Fig 8c), we mainly saw death across the three days. And finally, with our targeted therapy (Fig 8d), the differences between control cells and treated cells were really slight. It appeared that the mitosis is delayed and the behavior of each single cell was heterogeneous.

**Application of different simple parameters.** Here, we wanted to show the different application of parameters of Toto-Cell. Firstly, with the cladograms in mind, we wanted to look at the number of mitosis (Fig 9).

As expected we observed differences with cisplatin and doxorubicin conditions but not with AZD5363 condition. In term of quality of mitosis, we looked at the number of normal and abnormal mitosis. What we observed was that there was no significant differences. But one parameter that was really interesting was that we could see in doxorubicin condition that the few mitosis are abnormal. For the qualifying of the abnormal mitosis, we showed here for our cisplatin condition, that there was less non-dichotomous mitosis than in control condition, it may seem surprising but in fact there is less mitosis in total so it is normal that there is less non-dichotomous mitosis too. For the number of fusion, we observed no differences between our conditions, which means that there is no treatment that reduces the phenomena. Lastly for death, obviously we observed a difference between doxorubicin condition and control, we had more deaths and these deaths were without mitosis as presented in Fig 9.

**Application of different complex parameters.** We decided to look after a panel of complex parameters extracted from our data. In Fig 10 we showed the differences of time between the start of the video-microscopy and the first division. As expected we observed a difference for doxorubicin condition. In the same line, we investigated the differences of time between two mitosis, and again the only relevant condition was doxorubicin.

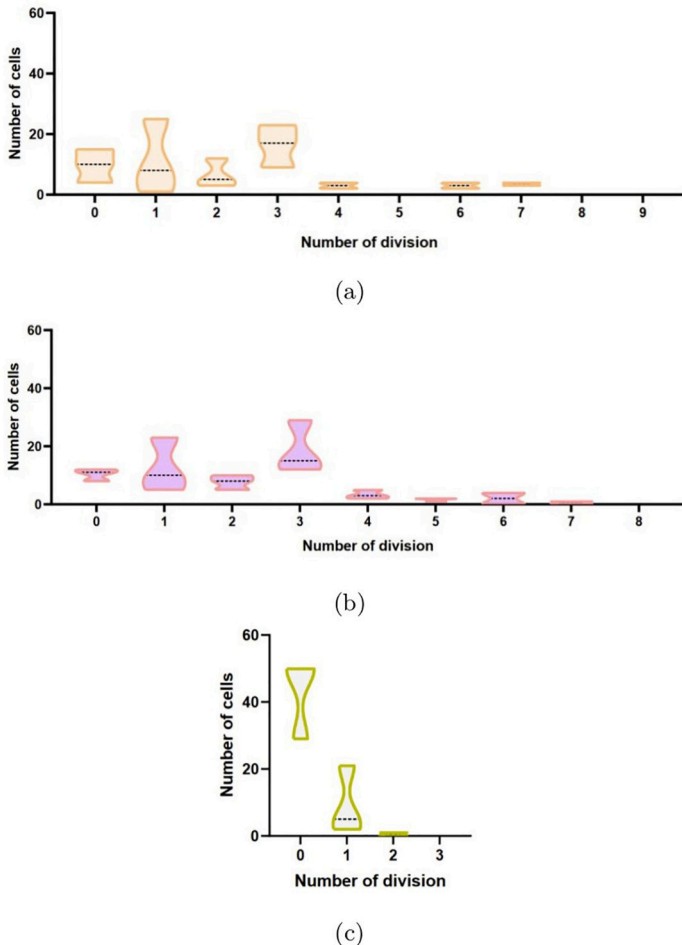

**Fig 11. Violin distribution of the number of mitosis (single cell following) from D1 to D3 of control and treated (24h) HEC-1-A cells.** Distribution's pattern of division's number for conditions, X-axis represents the number of division a cell did, Y-axis the number of cells that did the number of division(s). **(a)** Control condition, **(b)** AZD5363 condition ($20\mu M$), **(c)** Cisplatin condition ($10\mu M$) mean is represented by dot line, three independent experiments.

Following this study, we represented the different patterns of division in a single cell manner Fig 11: we represented here in violin diagrams, the number of cells that were doing a defined number of division(s). The AZD5363 and control condition have the same patterns, conversely to the cisplatin condition that did not go any further than two divisions

**Table 1. Representation of division's number per cell, HEC-1-A cells treated (24h) with doxorubicin ($3\mu M$).**

| Repetitions | Number of division per cell | | | | | | | |
|---|---|---|---|---|---|---|---|---|
| | **0** | **1** | **2** | **3** | **4** | **5** | **6** | **7** |
| n1 | 54 | 1 | 0 | 0 | 0 | 0 | 0 | 0 |
| n2 | 54 | 0 | 0 | 0 | 0 | 0 | 0 | 0 |
| n3 | 51 | 2 | 0 | 0 | 0 | 0 | 0 | 1 |

Each cell has made a precise number of division, here we represent for every repetition (n1, n2 and n3) the number of divisions each individual cell has experienced. This table showed that for doxorubicin's condition, there is so few division and it was impossible to represent in violin distribution.

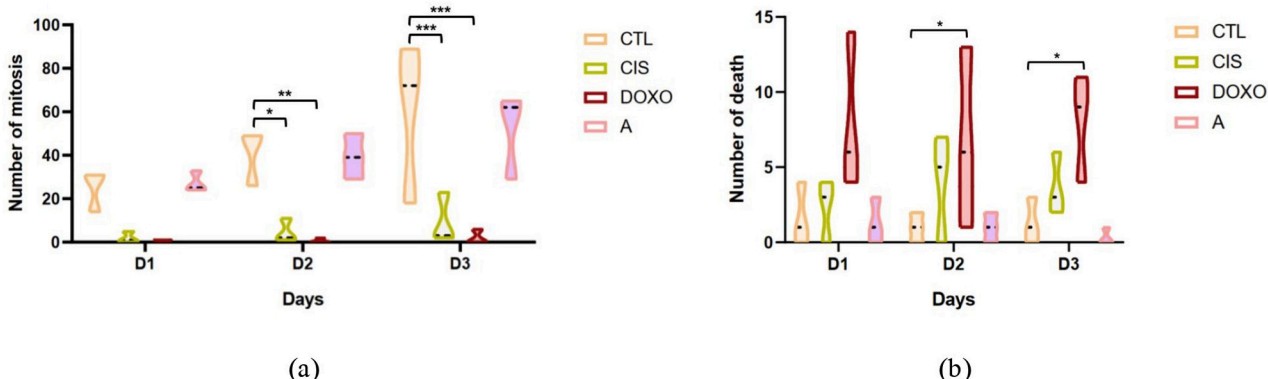

(a)    (b)

**Fig 12. Time-dependent study of mitosis and death events of control and treated (24h) HEC-1-A cells.** CTL: control condition, CIS: cisplatin condition ($10\mu M$), DOXO: Doxorubicin condition ($3\mu M$), A: AZD5363 condition ($20\mu M$). **(a)** Study of the number of mitosis between days, **(b)** Study of the number of death between days. Anova two ways, mean is represented by dot line, three independent experiments, *=$P < 0.05$; **=$P < 0.01$ and ***=$P < 0.001$.

into 3 days. For doxorubicin (see Table 1) there was so few divisions that it was not possible to represent them in that type of graph.

Ultimately, we wanted to study if there is any difference for division or death between days. We showed here that there is less division in day two and three for our chemotherapy conditions compared to the control. And more deaths for the doxorubicin condition on day 2 and 3 (Fig 12).

## Discussion

We showed here a new tool to help the manual analysis of video-microscopy. Toto-Cell software is very simple to use and can help to explain different results. Here we showed different parameters that can be interesting in cancer biology. We treated our cells with different drugs that are currently on the market. What we observed is that our HEC-1-A cells react differently between different conditions: cisplatin had few divisions but no death, doxorubicin either death or abnormal mitosis, and the effect of AZD5363 seems to lose its efficiency over time. It is really important to obtain these analysis because they can explain the results that we can have in other tests (such as clonogenic assays) and how a treatment is working or failing in a cell population. Taking the example of cisplatin, the fact that there is no mitosis during the first three days can explain the resistant phenotype of the HEC-1-A cell line, the cells are probably dividing again after the 3rd day. For doxorubicin, the fact that the mitosis are mostly abnormal can lead to the emergence of a new resistant tumor type.

The next important point is that in all of our graphs we observed heterogeneity. With at least 150 cells per condition, the most important point standing out was that the cells all have a different behavior even in the same well and in the same field. It is for this reason that single cell analysis is really important. That is why we developed a counter of events per cell and moreover we had a tracking back helper in Toto-Cell. The user, in the .txt file can point out with an icon a cell that had a different behavior and see on its cladogram how many cells had this same behavior 13.

For our experiments all the parameters were not relevant, for example the time between the start of our video and the first division, because we are treating our cell 24.5 hours before the video-microscopy. But in cases where the experimenter wants to see the direct effect of the drug or the therapy on the cells, by letting the drug into the media or in case of

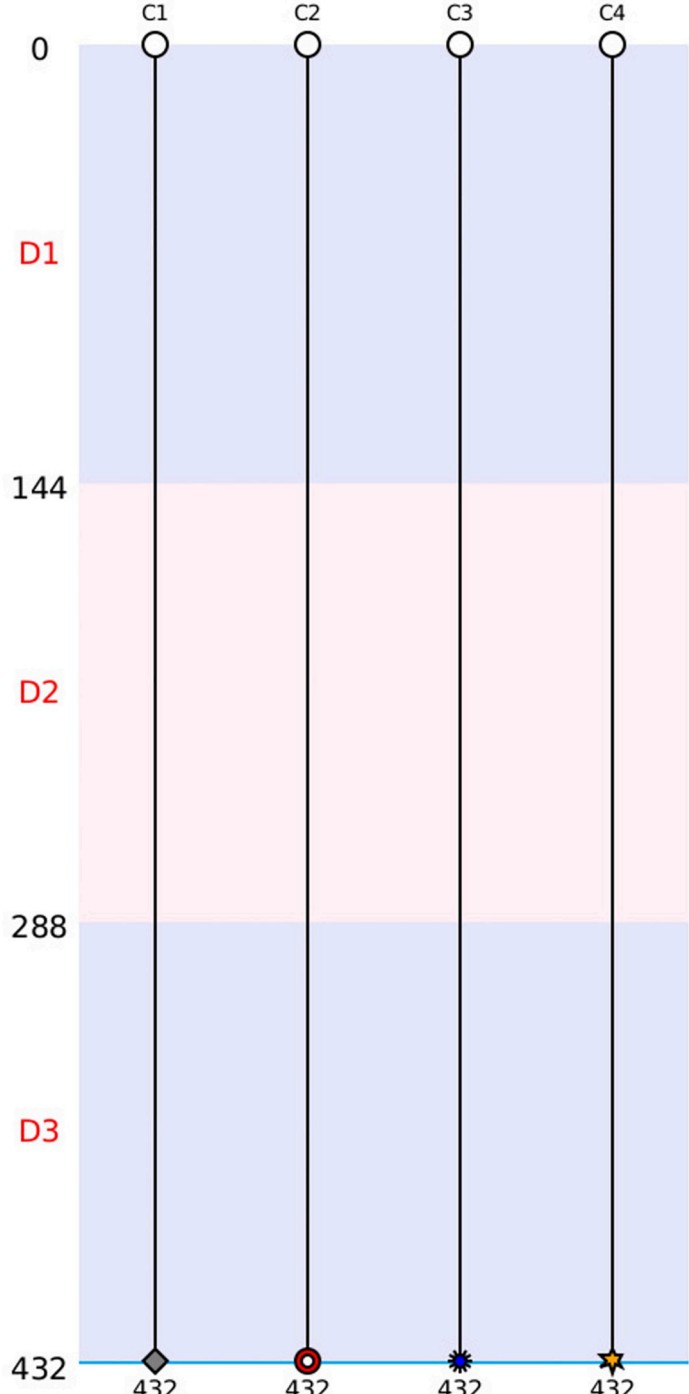

**Fig 13. Representation of tracking back cells.** The experimenter can choose between different forms and colors to point out a particular cell.

irradiating the cells, it is a very important pointing device. We know that all our parameters can be used in other treatment settings such as radiotherapy where mitotic death is such a great subject [12].

We are also very interested in using Toto-Cell in analyses other than cancer cells, as the software can be adapted very easily with as many parameters as the user wishes. For example, it would be interesting to use it with a model of C.Elegans, study of muscle cells that fuse over time, or with normal reproducing cells. It should be noted that this software can be made compatible with the output of existing analysis software, or even integrated with it if it is open-source (e.g. Trackmate [13]). This has not yet been done due to the current inability of such software to detect and/or follow the complex behavior of cancer cells (abnormal mitoses, fusions), which requires us to continue our analysis visually.

## Conclusion

Toto-Cell can be used regardless of: the type of cell, the imaging technology used, how many pictures the experimenter has, time between the frames or the number of cells tracked. It facilitates the manual analysis of video-microscopy and has several advantages:

- It is easy to use, and it doesn't require to have any IT skills, particularly through the web interface that has been developed;

- It helps speed up manual analysis of video-microscopy;

- It is possible to calculate new parameters that were difficult manually;

- It produces automatically cladograms with as much cells as the experimenter wishes;

- It is very simple to upgrade Toto-Cell with different and designed parameters.

As presented in the material and methods sub-section, Toto-Cell is only working with a .txt file. The experimenter has to follow a precise syntax. We provide a handbook of this syntax with example both on the sofware repository and on the web interface.

## Acknowledgments

The authors would like to thank Pascal Adam and Amélie Lemire for their help in proofreading and editing this article. A particular thank to Mélodie B. Plourde for her help in confocal microscopy.

## Author Contributions

**Conceptualization:** Eric Asselin.

**Data curation:** Léa-Isabelle Renaud, Christophe Renaud.

**Formal analysis:** Léa-Isabelle Renaud, Eric Asselin.

**Funding acquisition:** Eric Asselin.

**Methodology:** Léa-Isabelle Renaud, Eric Asselin.

**Project administration:** Eric Asselin.

**Resources:** Eric Asselin.

**Software:** Christophe Renaud, Samuel Delepoulle.

**Supervision:** Eric Asselin.

**Validation:** Léa-Isabelle Renaud, Samuel Delepoulle, Eric Asselin.

**Writing – original draft:** Léa-Isabelle Renaud.

**Writing – review & editing:** Léa-Isabelle Renaud, Christophe Renaud, Samuel Delepoulle, Eric Asselin.

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
