## [Decision Letter · Decision Letter 0]

1 Mar 2024

PONE-D-24-05115Toto-Cell : a new software to analyze cellular events during video-microscopyPLOS ONE

Dear Dr. Asselin,

Thank you for submitting your manuscript to PLOS ONE. After careful consideration, we feel that it has merit but does not fully meet PLOS ONE’s publication criteria as it currently stands. Therefore, we invite you to submit a revised version of the manuscript that addresses the points raised during the review process. Please submit your revised manuscript by Apr 15 2024 11:59PM. If you will need more time than this to complete your revisions, please reply to this message or contact the journal office at plosone@plos.org. Please include the following items when submitting your revised manuscript:A rebuttal letter that responds to each point raised by the academic editor and reviewer(s). You should upload this letter as a separate file labeled 'Response to Reviewers'.A marked-up copy of your manuscript that highlights changes made to the original version. You should upload this as a separate file labeled 'Revised Manuscript with Track Changes'.An unmarked version of your revised paper without tracked changes. You should upload this as a separate file labeled 'Manuscript'.

We look forward to receiving your revised manuscript.

Kind regards,

Alexander F. Palazzo, Ph.D.

Academic Editor

PLOS ONE

Journal Requirements:

   "The authors would like to thank Pascal Adam for his help in proofreading and editing this article. This work was supported by the Natural Sciences and Engineering Research Council of Canada (NSERC) (2019-06151 to EA)"

   "This work was supported by the Natural Sciences and Engineering Research280Council of Canada (NSERC) (2019-06151 to EA)."

   "This work was supported by the Natural Sciences and Engineering Research280Council of Canada (NSERC) (2019-06151 to EA)."

Additional Editor Comments:

As you can see, both reviewers found that the analysis pipeline was somewhat useful, but that this could be improved if it did not require a manually created input file, but instead could interpret output-files of commonly used imaging software. If the authors could address this issue, it would significantly strengthen the manuscript.

Reviewers' comments:

Reviewer's Responses to Questions

**Comments to the Author**

1. Is the manuscript technically sound, and do the data support the conclusions?

Reviewer #1: Yes

Reviewer #2: Yes

2. Has the statistical analysis been performed appropriately and rigorously? 

Reviewer #1: Yes

Reviewer #2: Yes

3. Have the authors made all data underlying the findings in their manuscript fully available?

Reviewer #1: No

Reviewer #2: Yes

4. Is the manuscript presented in an intelligible fashion and written in standard English?

Reviewer #1: Yes

Reviewer #2: Yes

5. Review Comments to the Author

Reviewer #1: In the present manuscript, Renaud et al. present an analysis pipeline for downstream processing of manually analyzed live-cell imaging data. In essence, the tool takes a table that, in a defined syntax contains, information about times of cell appearances, cell divisions, cell deaths, etc., and provides the user with a graphical representation of the lineage as well as several metrics that can be calculated from those data. A key feature is that it allows for division into more than two cells as well as cell fusion.

While I can see how this can be useful for users, the scope of the software is rather narrow. As is, I am not sure it warrants a full article, maybe some sort of micropublication (or simply release on github) would be sufficient.

As is, Toto-Cell requires input of a manually created 'data file'. I agree that analysis of live-cell imaging data typically requires some manual efforts (which often is the bottleneck for data analysis), but recent years saw a dramatic improvement on (semi)automated methdos for cell segmentation, tracking, and classifications. I think Toto-Cell could gain significantly in scope if the authors made an effort to make it compatible with the output-files of commonly used tools for cell tracking.

Reviewer #2: The article from Renaud et al propose a tool to analyze cell behavior after time-lapse imaging. It is available both as a stand-alone and web-based solution. It can retrieve many parameters like cell divisions, normal and aberrant, cell death, cells fusions, from a text file containing the different events over time. One downside is the manual tracking that have to be made on the time-lapse movies, a tedious part. Did the authors tried to use a solution like Trackmate to retrieve cells behavior and convert into a text file that can be used in the Toto-Cell solution?

The svg file should contain besides the dots and the time of the event, the name of the cell created during the process.

One analysis that could be interested to be done is for muscle cells that fuse over time. It might broaden the impact of the paper to another field.

6. PLOS authors have the option to publish the peer review history of their article (what does this mean?). If published, this will include your full peer review and any attached files.

Reviewer #1: No

Reviewer #2: No

---

## [Author Response · Author response to Decision Letter 0]

19 Mar 2024

*****Please see attached file for response to reviewers ******

March 18, 2024

Dear Dr. Alexander F. Palazzo,

To begin with, we would like to thank the editor and reviewers for their time spent re-viewing our manuscript. Their constructive criticisms are crucial to ensure robust demonstration and quality publication that is relevant for the research field. After care-ful consideration, we feel like we can address the vast majority of the comments and we appreciate the privilege we have to bring some clarifications where it was under-lined by the reviewers (responses are in blue).

It should be noted that all modifications to the manuscript are indicated in red in the revised version.

2. "The authors would like to thank Pascal Adam for his help in proofreading and editing this article. This work was supported by the Natural Sciences and Engineering Research Council of Canada (NSERC) (2019-06151 to EA)" We note that you have provided funding information that is not currently declared in your Funding Statement. However, funding information should not appear in the Acknowledgments section or other areas of your manuscript. [...]

Thanks for the information. We have removed this information in the acknowledgement section and please include funding statement:

This work was supported by the Natural Sciences and Engineering Research Council of Canada (NSERC) (2019-06151 to EA).

 "This work was supported by the Natural Sciences and Engineering Research280Council of Canada (NSERC) (2019-06151 to EA)."Please state what role the funders took in the study. If the funders had no role, please state: "The funders had no role in study design, data collec-tion and analysis, decision to publish, or preparation of the manuscript." 

Yes please, amend the role of the funder: 

This work was supported by the Natural Sciences and Engineering Research Council of Cana-da (NSERC) (2019-06151 to EA). The funders had no role in study design, data collection and analysis, decision to publish, or preparation of the manuscript.

4. We note that you have included the phrase “data not shown” in your manuscript. Unfor-tunately, this does not meet our data sharing requirements. PLOS does not permit refer-ences to inaccessible data. We require that authors provide all relevant data within the paper, Supporting Information files, or in an acceptable, public repository. Please add a citation to support this phrase or upload the data that corresponds with these findings to a stable repository (such as Figshare or Dryad) and provide and URLs, DOIs, or accession num-bers that may be used to access these data. Or, if the data are not a core part of the re-search being presented in your study, we ask that you remove the phrase that refers to these data.

We thank the editor for underlining this fact, we changed the sentence into our section “Application of different complex parameters”. We added a table showing the different numbers of mitosis in the doxorubicin condition. We apologize for not meeting the data sharing requirements of PLOS.

Editor Comments:

As you can see, both reviewers found that the analysis pipeline was somewhat useful, but that this could be improved if it did not require a manually created input file, but instead could interpret output-files of commonly used imaging software. If the authors could address this issue, it would significantly strengthen the manuscript.

We would like to thank the editor for his comment on our manuscript. We agree with the reviewers that having output files coming from imaging software like Trackmate could be useful and improve the value of our manuscript. Unfortunately, as we will ex-plain in the section “answer to the reviewers”, Trackmate do not recognise our cells perfectly nor the fusion events which do not allow us to use this software. 

Comments to the authors:

Reviewer #1: In the present manuscript, Renaud et al. present an analysis pipeline for downstream processing of manually analyzed live-cell imaging data. In essence, the tool takes a table that, in a defined syntax contains, information about times of cell appearances, cell divisions, cell deaths, etc., and provides the user with a graphical rep-resentation of the lineage as well as several metrics that can be calculated from those data. A key feature is that it allows for division into more than two cells as well as cell fusion.

While I can see how this can be useful for users, the scope of the software is rather narrow. As is, I am not sure it warrants a full article, maybe some sort of micropublica-tion (or simply release on github) would be sufficient.

We would like to thank the reviewer for mentioning his enthusiasm in our finding and manuscript. We believed there is enough material into the article to be worth publish-ing in a scientific journal. Moreover, we think that Toto-Cell could be used in many fields of research and that is why we think it merits larger visibility.

As is, Toto-Cell requires input of a manually created 'data file'. I agree that analysis of live-cell imaging data typically requires some manual efforts (which often is the bottle-neck for data analysis), but recent years saw a dramatic improvement on (semi)automated methdos for cell segmentation, tracking, and classifications. I think Toto-Cell could gain significantly in scope if the authors made an effort to make it com-patible with the output-files of commonly used tools for cell tracking.

We agree with the reviewer that the automatic analysis of video-microscopy sequences is a fast-growing field, whether using traditional segmentation techniques or more re-cent techniques derived from deep learning. Nevertheless, the tests we carried out with available software (e.g. Trackmate) were not conclusive on our sequences. Complex events such as abnormal mitoses or fusions are not detected by this type of software, leading to large amounts of erroneous results. This is the reason why we have chosen to continue with a manual analysis of our sequences and developed Toto-cell to help visualize and analyze our results. It should be noted that this software can be made compatible with the output of existing analysis software, or even integrated into it if it is open-source (e.g. Trackmate again), but this has not yet been done for the reasons given above. This point has been added in the introduction part of MS and at the end of the discussion.

Reviewer #2: The article from Renaud et al propose a tool to analyze cell behavior after time-lapse imaging. It is available both as a stand-alone and web-based solution. It can retrieve many parameters like cell divisions, normal and aberrant, cell death, cells fu-sions, from a text file containing the different events over time. 

One downside is the manual tracking that have to be made on the time-lapse movies, a tedious part. Did the authors tried to use a solution like Trackmate to retrieve cells be-havior and convert into a text file that can be used in the Toto-Cell solution?

We thank the reviewer for his thorough evaluation of our manuscript and for providing comments. TrackMate offers customizable settings for adapting cell tracking to the specific characteristics of the videos we wish to analyze. However, these settings are ineffective for our particular type of cells. The primary determinant for achieving accu-rate automated analysis lies in the choice of detector. Options include the DoG detec-tor, Thresholding detector, Hessian detector, Label image detector, LoG detector, Man-ual annotation, and Mask detector. Despite experimenting with various detectors, none were efficient in reliably detecting and tracking cells within our videos. Although we were initially able to fine-tune filters for individual frames, maintaining consistent sig-nal across successive frames was impossible. This renders the cell tracking functionality inadequate for our video analysis needs. It is also detecting cells where there are none. This inaccurate tracking results in the creation of false segments in the cladogram pro-duced by Trackmate. Moreover, after wide experimentation with all available settings across our videos, TrackMate has consistently failed to detect cell fusion events. Cell fusion is prevalent in cancerous cells and represents a common yet critical event to identify in our study. 

The svg file should contain besides the dots and the time of the event, the name of the cell created during the process.

Adding the name of the cell next to each point representing a cell is not a difficulty in itself, but can pose a problem of readability when complex series of events occur. The image below illustrates this problem.

The name of each cell has been written above it, with the time of the event moved to the right of each 'complex' event (mitosis, fusion). For cell 1, everything is legible. But for cell C5, which highlights a complex series of events, it is difficult to read both the names of the cells and the time of the event. Cell C5 highlights another problem when cell names become long due to multiple divisions. There is no 'right' way to write cell names and ensure that they remain legible regardless of the complexity of the events that may occur. This is why we have chosen not to write the cell names except for the initial names, which are important for discerning the cells in the videos. 

One analysis that could be interested to be done is for muscle cells that fuse over time. It might broaden the impact of the paper to another field.

Concerning the study of muscle cells, it was outside the scope of this manuscript and outside our area of expertise. We wanted to focus solely on cancer cells for this first demonstration of proof-of-concept. However, we agree with the reviewer that it could be really relevant to do this kind of study. We have added in the discussion section, the idea of studying muscle cells. Our software is Open-Source and we remain at disposal for any future collaborations in any field of research and if we need to change or add any parameters to study this type of cells, we will be happy to do so.

---

## [Editor Report · Decision Letter 1]

28 Mar 2024

Toto-Cell : a new software to analyze cellular events during video-microscopy

PONE-D-24-05115R1

Dear Dr. Asselin,

We’re pleased to inform you that your manuscript has been judged scientifically suitable for publication and will be formally accepted for publication once it meets all outstanding technical requirements.

Kind regards,

Alexander F. Palazzo, Ph.D.

Academic Editor

PLOS ONE
---

## [Editor Report · Acceptance letter]

30 Apr 2024

PONE-D-24-05115R1 

PLOS ONE

Dear Dr. Asselin, 

I'm pleased to inform you that your manuscript has been deemed suitable for publication in PLOS ONE. Congratulations! Your manuscript is now being handed over to our production team.

Kind regards, 

on behalf of

Dr. Alexander F. Palazzo 

Academic Editor

PLOS ONE